# Construction of an Automated Removal Robot for the Natural Drying of Cacao Beans

**DOI:** 10.3390/s25051520

**Published:** 2025-02-28

**Authors:** José Tuanama-Aguilar, Carlos Ríos-López, José Luis Pasquel-Reátegui, Carlos Rodríguez-Grández, John C. Santa-Maria, Janina Cotrina-Linares, Cristian García-Estrella, Felix-Armando Fermin-Perez

**Affiliations:** 1Facultad de Ingeniería de Sistemas e Informática, Universidad Nacional de San Martín, Tarapoto 22000, Peru; jjtuanama@unsm.edu.pe (J.T.-A.); carios@unsm.edu.pe (C.R.-L.); crodriguez@unsm.edu.pe (C.R.-G.); jsantamaria@unsm.edu.pe (J.C.S.-M.); jcotrina@unsm.edu.pe (J.C.-L.); cgarcia@unsm.edu.pe (C.G.-E.); 2Grupo de Investigación en Ingeniería y Tecnología Agroindustrial, Facultad de Ingeniería Agroindustrial, Universidad Nacional de San Martín, Tarapoto 22000, Peru; 3Laboratorio de Robótica e Internet de las Cosas, Facultad de Ingeniería de Sistemas e Informática, Universidad Nacional Mayor de San Marcos, Lima 15081, Peru; fferminp@unmsm.edu.pe

**Keywords:** robotics, automation, postharvest, *Theobroma cacao* L., control system, optimization

## Abstract

Cacao producers often obtain low-quality beans due to the poor manual drying process. This study proposes the construction of an automated prototype robot for the removal during natural drying of cacao beans at Cooperativa Agraria Allima Cacao Ltd., Peru, and evaluates its effectiveness on bean quality. The robot comprises three modules (control, displacement, and removal) and motion sensors. Four 400 kg batches were analyzed, obtaining moisture contents of 6.71, 8.59, 7.74, and 6.80% with the robot, compared to 7.86, 7.94, 6.60, and 7.54% with the manual method. The standard deviations in the grains treated with the robot were lower, evidencing a more uniform drying. In addition, the total phenol content was higher in the robot-dried beans, indicating better preservation of bioactive compounds. Although the robot did not reduce the drying time, we conclude that its sustainable and economically accessible design contributed to obtaining higher-quality beans compared to the conventional method, with more homogeneous drying and better preservation of phenols, key aspects for producing high-quality cacao.

## 1. Introduction

Since the Mayan civilization in Central America (400 AD), cacao (*Theobroma cacao* L.) has been considered a source of food for human consumption [1]. Studies highlight its abundance of polyphenols, particularly flavanols, which benefit human health with antioxidant, anti-cancer, anti-diabetic, anti-inflammatory, anti-obesity, and anti-allergy properties [2,3,4]. According to [5], cacao is a rich source of fiber (40–26%), lipids (24–10%), proteins (20–15%), and carbohydrates (15%); it also contains a small amount of micronutrients (<2%), including minerals such as phosphorus, calcium, potassium, sodium, magnesium, zinc, and copper, as well as vitamins A, B, and E.

Cacao is mainly cultivated on a large scale in areas with hot and humid climates in countries such as Ecuador, Colombia, Venezuela, Brazil, Peru, Bolivia, Trinidad and Tobago, and Mexico, being a source of income for agricultural families [6,7]. The cacao tree begins to be productive after three years in favorable conditions and can produce pods throughout the year, depending on the area and altitude of cultivation. From five years onwards, it is considered to achieve its full performance, with a useful life of up to 100 years. However, its optimal productivity period is approximately 40 years [8,9].

In Peru, one of the leading producers and suppliers of fine and aromatic cacao and the world’s second-largest producer of organic cacao, national production grew at an annual average of 10.1%, made up of 53.3% of the Trinitario variety (Junín), 37.3% Amazonian outsider (Cusco and Ayacucho), and 9.4% creole (northern San Martín, Amazonas, and Cajamarca). In the first semester of 2023, cacao production decreased by 4.8% compared to the same period in 2022, reaching a volume of 86.2 thousand tons, a reduction attributed to lower production in the central producing regions, such as San Martín, Junín, Huánuco, and Cusco [10].

Regarding the drop in Peruvian cacao production, refs. [11,12] point out that it is due to deficiencies in the value chain, particularly in the fermentation and drying processes, since the use of jute or raffia bags for weighing on transportation is carried out with second- or third-use sacks [13]. Subsequently, the selection and handling operations of the cacao bean require more excellent drying [14], evidencing that low- and medium-scale producers need to become more familiar with the protocols and standards for post-harvest handling of cacao. They are negotiating only with the maximum moisture value of the dry grain (7.5%).

Likewise, ref. [11] reveals that during the drying of the cacao bean, 8% humidity is reached. Still, in most cases, it is marketed with a humidity of 12 to 16% in the San Martín region, putting at risk biological contamination of the grain due to possible fungal contamination and the production of toxins, such as Ochratoxin A (OTA), which are a concern in the food industry due to their toxic effects on human health, such as nephrotoxicity (kidney damage) and potential carcinogenic effects [15,16,17].

Consequently, the precarious process of drying the cacao bean due to the need for more technology results in inappropriate humidity levels that prevent uniform drying [18]. This deficiency compromises the quality of the cacao, reducing its value in the market and increasing the risk of contamination that affects product safety [19,20]. We recognize that low investment in infrastructure and training in post-harvest practices highlights these problems, generating labor overload, low competitiveness in national and international markets, and restricting the ability to obtain a fair price.

Drying cacao beans under direct sunlight (natural drying) is one of the oldest and most predominant techniques due to its simplicity and low cost [21]. Still, it is difficult to control, given the climatic conditions and labor often required, leading to production losses due to uniform drying [22]. Currently, forced-convection-drying solutions have been proposed to allow the production of heterogeneous batches with constant good quality in a short time [23,24,25]. However, most of these proposals were designed to use fossil fuels as the main source of heat, with limitations associated with the great potential for pollution, high cost, and finite reserves.

Therefore, the objective of this research was to build a prototype of an automated removal robot for natural cacao bean drying that is sustainable and economically profitable. We also evaluated the effect of automated stripping on cacao bean quality to ensure its adoption and scalability in the market.

## 2. Materials and Methods

### 2.1. Intervention Area

The study was carried out in the agro-industrial plant of the Cooperativa Agraria Allima Cacao Ltd. (18M 374 048 m E, 9 272 555 S, 190 m.a.s.l), located in the Chazuta district, province, and San Martín, Peru, department. This area has temperatures between 19.6° and 36.6 °C and a relative humidity between 75.3% and 81.5%. The cooperative has drying chambers that protect the grains from rain, provide shade, and let in natural light and wind. We use the cacao-bean-drying chamber equipped by the cooperative to guarantee a drying process that does not exceed 50 °C temperature or 70% humidity.

### 2.2. Assembly of the Robot—Prototype

The automated non-conventional removal robot comprises three modules: control module, displacement module, and removal module, which simulate the traditional method of manual or conventional removal applied in the cacao-bean-drying process at the company. Below, we describe the robot assembly phases.

#### 2.2.1. Three-Dimensional Modeling

The 3D modeling of the robot (Figure 1) was carried out using the free software Blender (version 3.2.1), a widely used tool in design and simulation due to its versatility. This modeling allowed us to detail the structural and functional characteristics of the robot, ensuring it meets the specific requirements for the automatic removal of cacao beans during natural drying. Figure 1a presents an isometric view of the model, highlighting the robot’s main components, such as its movement system and the elements responsible for bean removal. Figure 1b offers a general view of the drying scenario, integrating the robot with the work area and surroundings. Figure 1c provides a top view, showcasing the arrangement of the internal mechanisms and their interaction with the workspace. The implemented modules are described in the following sections.

#### 2.2.2. Control Module

The control module (Figure 2) includes a controller programmed in ladder language, responsible for driving the 1.5 HP reduction motor and directing the robot’s operations. This controller connects to a display that provides a visual interface for monitoring and adjusting parameters. We incorporate a communications switch that allows system interaction and control to facilitate communication. A 24-volt power supply guarantees the robot’s continuous power supply. This supply supplies the necessary energy for the control module devices, which are protected by 16 A fuses to prevent overloads.

A 10 kΩ potentiometer allows you to adjust the speed parameters in the frequency converter, regulating the movement of the removal module on the drying bed according to the specific needs of removing cacao beans. The system also has a relay that activates an acoustic signal of the robot’s movement. In addition, we incorporate position sensors in the drying bed, one at the beginning and another at the end, allowing the controller to know the location of the removal module.

#### 2.2.3. Displacement Module

The displacement module comprises a 1.5 kW motor connected to a reducing box with an 80:1 ratio. This system, utilizing shafts with bearings linked to a drag chain 428H-150L ST, enables the removal support to move forward and backward based on the motor’s rotation direction (Figure 3).

#### 2.2.4. Removal Module

The removal module comprises 57 fingers distributed in two batteries: one with 29 fingers and another with 28. These batteries are organized neatly within a wooden structure. The fingers have a vertical movement that allows them to adapt to the surface of the drying bed. In addition, each finger has stops at the bottom and top to limit travel. It should be noted that the removal module is made entirely of wood, which has been previously treated by bathing it with the cacao juices obtained during the fermentation stage, guaranteeing its resistance. The time between each removal cycle is configured through the display on the control module (Figure 4).

For the electrical power supply, twelve TrinaSolar TSM-455DE17M solar panels, each 455 W, a Growatt SPF 5000 ES 48 V 100 A inverter charger, which is a pure sine wave inverter characterized by having a maximum output power of 5000 W, and three Shoto SDA10-4850 48 V 50 AH 2.4 Kwh Server Rack LiFePO4 batteries were used. Table 1 details the model and manufacturer of each hardware component used in the robot’s construction.

In summary, a robot was designed to automatically stir cacao beans in a natural drying chamber, reducing their humidity to 7%. The robot employs a rail-based movement system to traverse a horizontal drying table, removing the cacao beans. Its modular design enables it to adapt to various batches of cacao beans, offering flexibility to accommodate production variations. It features two sets of wooden paddles with vertical movement that conform to the surface of the drying bed. The cycles for removing cacao beans can be easily programmed via a screen on the control module. The linear displacement robot operates periodically, 24 h a day, 7 days a week, as programmed by the user.

The robot includes a program in the PLC that manages the following logical sequence: (a) The motor turns on after a waiting time and stays on, turning the shaft connected to chains on the sides. This makes the robot move towards the end of the drying bed until one of the sensors (switch sensor) located at the end of the bed causes the robot to stop and then reverse the motor’s rotation to make the robot go to its initial position, this rotation will stop the motor when one of the sensors (switch sensor) located at the beginning of the bed detects the arrival of the motor. (b) The process described above is cyclical, and the waiting time for removal can vary at any point during the process. (c) The process concludes when the cacao beans reach a humidity of 7%, unless an emergency arises or the user decides otherwise.

### 2.3. Effectiveness Test

To evaluate the effectiveness of the automated non-conventional removal robot on the natural drying of the cacao bean, we consider the following conditions:

#### 2.3.1. Weather Conditions

The tests were conducted with temperature (maximum 59.3 °C and minimum 24.1 °C) and relative humidity (maximum 92% and minimum 15%). Also, light inflows (maximum 50,910 lx and minimum 0.28 lx) and wind conditions (0.92 K/h) were recorded. The automated removal robot (non-conventional) and method of natural drying of the cacao beans (conventional drying) were installed in a drying chamber that protects the beans from rain. Table 2 shows the climatic conditions observed during the experiments.

#### 2.3.2. Cacao Beans

The cacao beans were provided by the Cooperative Agrarian Allima Cacao from November 2023 to January 2024 (a wet season characterized by an increase in precipitation). We took samples from three parts: entry, half, and fund (Figure 5), every day of the drying process until a moisture content of 6–8% was achieved (7–10 days).

#### 2.3.3. Reagents

Sodium carbonate (CAS 144-55-8) and a Folin–Ciocalteu reagent were purchased from CDH—Central Drug House (New Delhi, India). Anhydrous gallic acid (CAS 149-91-7) was purchased from Merck KGaA (Darmstadt, Germany). We used ethyl alcohol (96°). We obtained the water in the experimental trials using distillation equipment (Model 2008, GFL, Burgwedel, Germany).

#### 2.3.4. Determination of Moisture Content

We determined humidity using method 930.04 of the Association of Official Analytical Chemists (AOAC). In a Petri plate, we weighed approximately 5 g of whole cacao beans, and they were subsequently placed in a conventional oven (Venticell 111—ECO line, BMT medical, Kassel, Germany) at 105 °C until constant weight was obtained (24 h). To calculate the results, we used Equation (1):(1)$$Moisture\;\left(\%\right)=\frac{W_{1}-W_{2}}{W_{1}}\times 100\;$$
where *W*_1_ = wet grain weight, and *W*_2_ = dry grain weight. Weights were determined with an analytical balance (Pioneer TM PX224, OHAUS, Miguel hidalgo, Mexico).

#### 2.3.5. Phenolic Compound Extraction

It was performed using ultrasound-assisted extraction (UAE). We carried out the extraction with the testa (external part) and cotyledon (internal part) of the cacao beans before and after drying. To do this, we manually separate the testa from the cotyledon, then grind it in a mortar to reduce the particle size and increase mass transfer. The extraction was performed with a tip ultrasound (900 W of power, 12.5 mm de Ø, frequency of 20 kHz, Biobase, Jinan, China), adding 0.5 g of the sample with 10 mL of the solvent (ethanol). Each treatment was subjected to 40% power for 6 min. Then, it was taken to a centrifuge (EPPENDORF, 5702, Hamburg, Germany) at 3500 rpm for 5 min. We recovered the supernatant and stored it in amber bottles at 2 °C until subsequent characterization.

#### 2.3.6. Determination of Total Phenolic Content

We determined the content of phenolic compounds (TPC) of the extracts using the Folin–Ciocalteu spectrophotometric method proposed by [26] with certain modifications. We mixed 200 μL of the ethanolic extract, 200 μL of the diluted Folin–Ciocalteu reagent (1:2), 400 μL of sodium carbonate (10% *p*/*v*), and 3200 μL of distilled water. The mixture was shaken for 5 s and left incubated in the dark for 30 min at room temperature (25 °C). Absorbances were measured at 765 nm in a spectrophotometer (GENESYS 50, ThermoScientific, Madison, WI, USA). To prepare the calibration curve, we used a standard solution of gallic acid at different concentrations (0–0.12 mg/mL). Results were calculated with Equation (2):(2)$$TPC=\frac{c\times V}{m}\;$$
where “*c*” is the sample concentration in the calibration curve, “*V*” is the volume of the solvent used, and “*m*” is the weight of the sample used [27]. The results were expressed as mg gallic acid equivalents per g raw material (mg GAE/g MP).

#### 2.3.7. Determination of Grain Integrity

To evaluate the integrity of the cacao bean, we took a representative sample of 60 beans every 24 h, both from the conventional drying process and from drying using the automated removal prototype. The collected grains were packed in dark vacuum Ziploc bags and duly identified. Afterward, they were sent to the Process Engineering Laboratory of the National University of San Martín for the respective analysis. We determined the percentage of grain integrity (GI) through the following formula: GI = number of whole grains/total number of grains.

#### 2.3.8. Determination of Energy Yield

We evaluated the energy performance through energy consumption of the automated cacao bean non-conventional removal robot using a power consumption meter built into the control panel. We carried out the initial and final recording of the energy consumed until we achieved a humidity percentage of 7%.

#### 2.3.9. Determination of the Removal Cycle

The prototype “robot” makes a round trip on the drying table in 8 min. In the validation process, we consider a route trip every 60 min. We calculate the removal cycles once a humidity percentage of 7% is obtained.

### 2.4. Analysis of Data

We had two experimental and three analytical repetitions to determine the humidity and total phenolic content. The results were expressed as mean ± standard deviation and evaluated using one-way analysis of variance (ANOVA), with a significance level of 5%. We used a Tukey’s test (α = 0.05) to compare the means. A statistical analysis was performed with MINITAB^®^ statistical software (version 18.1.0, Minitab Inc., State College, PA, USA). We generated the drying-curve graphs with Microsoft Office Excel 2019. This tool also helped us record data on grain integrity, energy yield, and removal cycle indicators.

## 3. Results and Discussion

### 3.1. Robot Functionality

We developed an automated removal robot (prototype) based on control, displacement, and sensor systems, which works with energy from solar panels. For its functionality, first, you configure the necessary parameters through the control screen, which include the waiting time to start the removal, the round-trip removal time, the waiting time after touching a sensor, and the engine’s status (on or off). Then, the waiting time accumulates until the condition is met before the removal can start. After removal, the engine’s condition is checked, and the process is continued if the engine is on; otherwise, the process ends.

The 1.5 HP reducer motor remains on, moving the robot toward the end of the drying bed. The wait time is accumulated after touching the bed end sensor until the condition is met. Then, round-trip removal time is accumulated until the robot touches the bed start sensor. If all conditions are met, the removal process continues; otherwise, the process terminates until the parameters are configured again.

To the best of our knowledge, there is no history of the development of automated removal robots for the natural drying of cacao beans with the modules and assembly design used in the present study since the vast majority of research, as we mentioned in the Introduction, have proposed drying solutions by forced or artificial convection that allow for producing heterogeneous batches with constant quality in a short time [23,24,25,28]. Still, most of these proposals were designed to use fossil fuels as the primary heat source, having associated limitations to the great potential for contamination, high cost, and finite reserves.

Therefore, our proposal for a prototype robot (non-conventional removal) is eco-friendly since it uses solar panels, reducing dependence on non-renewable energy sources and reducing the carbon footprint. In addition, the robot maintains a controlled natural drying, ensuring the continuity of the grain fermentation process, which is essential to preserve its quality and guarantee a fair price in the market. We also highlight that the total construction cost of the robot is 5000 dollars, making it accessible to small and medium-scale producers to improve the sustainability of their operations without incurring high expenses involving labor and drying control processes inefficient.

### 3.2. Grain Integrity, Energy Performance, and Removal Cycles of the Robot (Non-Conventional)

When evaluating the integrity of cacao beans using the robot (non-conventional removal), we observed that the integrity percentages were slightly lower than those of the conventional removal method. In batch I, the integrity of the grains was 98.8% with the robot, while the conventional (traditional) method showed an integrity of 99.8%. In Lot II, the integrity of the robot was 98.81%, while the conventional method reached 99.05%. These results indicate that, although the use of the robot tends to reduce the integrity of the cacao beans slightly, the difference is minimal. Therefore, we confirmed that non-conventional removal effectively removes grains without causing significant damage, showing promise for its use in industrial processes. Additionally, the robot’s additional advantages, such as automation and consistency in the drying process, can offset this minimal reduction in grain integrity.

On the other hand, in batches of 400 kg, the robot (non-conventional removal) consumed an average of 0.02 kW (beans with 6.8% humidity) to 0.06 kW (beans with 52.9% humidity) per cacao bean removal cycle. During the entire drying process, a minimum consumption of 3 kW and a maximum of 13 kW were recorded, according to the number of removal cycles carried out until reaching the relative humidity percentage (6–8%) to complete the drying process. This consumption range reflects the robot’s ability to adjust its operation according to the specific needs of each batch, optimizing the drying process. A minimum consumption of 3 kW suggests high efficiency under favorable conditions. In comparison, the maximum of 13 kW indicates the ability to handle more demanding situations, such as beans with higher humidity or adverse environmental conditions. Compared to conventional methods, the robot shows low energy efficiency and consistency in drying quality, being able to compete in effectiveness and human cost. Although consumption may vary, the robot’s ability to operate quickly and precisely can result in lower cooperating costs and a smaller environmental footprint. Automation ensures a more uniform process and reduces the risk of damage to the beans, which could improve the quality of the final product and increase sustainability in cacao production.

Regarding the removal cycles, the batch I recorded the maximum with 218 cycles in 10 days, reducing the beans’ humidity to 7%, demonstrating the robot’s ability to maintain a high frequency of removal when necessary, ensuring uniform drying. On the other hand, batch II reported the minimum number of cycles with 141, achieving a humidity reduction of 7%. The fewer cycles in batch II may indicate more favorable initial conditions, such as lower humidity or a more efficient drying environment. The variability in the removal cycles between batches is due to the natural drying process used in this study, which is greatly influenced by the beans’ initial conditions and the site’s climatic conditions. This highlights the robot’s ability to adapt to different drying conditions, adjusting its operation to optimize the efficiency and quality of the process. Table 3 shows the averages of drying time, energy consumption, removal cycles, and integrity of the cacao beans according to the method used.

### 3.3. Drying Kinetics

Fresh cacao beans have three layers: pulp, testa, and cotyledon, but the pulp degrades during fermentation [29]. After fermentation, the moisture in the grains is around 55%, and to store them safely, it must be reduced to 6–8% [30,31]. With humidity above 8%, the grain would be susceptible to mold development [32], while less than 5% would make the shell more brittle, increasing the risk of breakage during handling [33].

During drying, the testa’s hardening becomes brittle, while the cotyledons shrink, causing a decrease in grain size [34]. Figure 5 shows the humidity of the cacao beans during each day of the drying process for the four batches studied. We observed that the moisture loss is more significant at the beginning and then gradually decreases; this behavior in the drying kinetics has also been reported in other works [35]. The first stage can be attributed to the high surface moisture content of the testa, while, in the final phase, the diffusion of moisture from the cotyledon towards the outer surface is restricted by the hardening of the testa; that is, there is less permeability in the system [36].

Regarding the drying curves, the batches do not show uniform behavior. In batches I and II, the non-conventional removal line (robot) is slightly below the conventional line, while in batches III and IV, the opposite is accurate; however, a significant difference is not observed between the humidities of the grains treated with the robot and conventionally, which becomes more evident at the end of drying since they reach similar humidity levels.

On the other hand, the drying time was variable (between 7 and 10 days), which makes sense considering that the experimentation was carried out in the wet season, where weather conditions are unpredictable, and precipitation usually increases. Those batches that took longer were probably because, in rainy climates, the relative humidity increases, causing a lower drying speed [37]. This phenomenon became more evident in batch I, where the drying time was longer than in the others (10 days); there was even a slight increase in humidity from days 8 to 9 (Figure 6a).

The initial humidities of the grains observed for the non-conventional removal (robot) were 47.56, 41.85, 54.80, and 50.75%, and for the conventional removal, 50.80, 43.46, 53.91, and 55.13% in batches I, II, III, and IV, respectively. These values align with other research reports since fermented cacao beans usually contain between 55 and 60% moisture [38]. Values of 53.7% [31], between 59.15% and 60.37% [39], and around 49% [40] have been reported.

After drying, moisture contents of 6.71, 8.59, 7.74, and 6.80% were reached for non-conventional removal and 7.86, 7.94, 6.60, and 7.54% with conventional removal in batches I, II, III, and IV, respectively. Generally, the standard deviations regarding the moisture content of the grains where the robot was used for removal are lower than those observed in conventional removal, which could indicate that the robot allows for more uniform drying.

The initial and final moisture contents align with those reported in other sun-drying studies. Additionally, an initial moisture level of 48.42% was found in grains fermented for 6 days, which was reduced to 9.78% by the fourth day of drying [34]. They also predicted that open sun drying would take 6 days to achieve a humidity level of 5 to 6% [41]. Using the traditional sun-drying method, moisture decreased from approximately 60% to 6.83% and 7.67% over 6 days. Ref. [42] observed humidity levels of 55% to 60% in fermented grains, which, after traditional solar drying, reached between 6.83% and 7.67%. Ref. [40] compared various drying methods, including sun drying on an open patio, where the grains’ moisture content fell from 49.86% to 7.14% in 30 h. Ref. [43] reported that the grains had a moisture level of 52.18% after fermentation, and through traditional drying (direct sun exposure), it reached 7.76% in 7 days.

If drying takes several days, there is a chance of contamination by mold, which damages the grain [34]. However, when drying in the sun, environmental fluctuations (temperature and relative humidity) occur, which causes slower drying and, therefore, longer drying time [35]. It can take up to 22 days in rainy periods, and in dry seasons, it can last 7 days [31]. The moisture content of the grains and the drying time are influenced by multiple factors, such as the variety and the fermentation period [41,42], high relative humidity [44], and the loading level [45].

Although natural drying is a widely used method, and grains with lower acidity content and good flavor development are obtained compared to artificial methods, it requires much labor since the grains are regularly turned manually [22,46]. Implementing the robot (non-conventional removal) would facilitate this task and, in turn, allow for a more uniform drying of all the cacao beans.

### 3.4. Total Phenolic Content (TPC)

During fermentation, chemical changes in cacao beans continue during drying until the moisture content is below 7% or the enzymes are inactivated [47]. The polyphenols in cacao beans have many benefits, such as anti-cancer, anti-depressant, anti-hypertensive, anti-inflammatory properties, etc. [32]. In addition, phenolics in cacao are in a higher proportion and have a higher antioxidant activity than tea and red wine [48]. In cacao beans, three groups of phenolics are distinguished: proanthocyanidins (58%), catechins (37%), and anthocyanins (4%). Generally, they represent between 6 and 8% by weight of the dry fermented cacao bean [49] and are responsible for promoting an astringent flavor [32,50].

Table 4 shows the total phenolic content depending on the grain’s state (start and final), the type of removal (non-conventional and conventional), and the part of the grain (testa and cotyledon).

According to Tukey’s test, averages represented by different lowercase letters in the same column indicate a significant difference between grain states for the same level of removal type and grain part (*p* < 0.05). Averages marked with capital letters in the same row demonstrate a significant difference between the prototype robot and the conventional drying method under the same conditions and grain part (*p* < 0.05). Averages denoted by superscript letters in the same row signify a significant difference between the testa and cotyledon for the same level of removal type and grain state (*p* < 0.05).

In all cases, there was a significant decrease in the TPC at the beginning and the end of drying; this is due to the enzymatic browning caused by polyphenol oxidase and the non-enzymatic browning caused by the polymerization of quinones caused, in turn, by the accumulation of insoluble compounds [43]. The reduction in polyphenols depends on several factors. The degradation rate increases the higher the temperature and drying time [36]; more significant degradation has also been seen under conditions of high relative humidity [47]; also, due to the polarity of polyphenols with water, these dissolve and are subsequently dragged to the surface. It was observed [51,52] that the most significant loss of polyphenols during drying occurs in the first three days due to the more substantial presence of humidity, which improves the polyphenol oxidase activity.

To compare the degradation of the TPC between non-conventional drying (robot) and conventional drying, we can take batch II as a reference since it is the only one where the amounts of phenols are statistically equal at the beginning, in addition to having the shortest drying time (7 days). We can note that the initial TPCs in the testa were 4.33 and 4.23 mg GAE/g MP and decreased to 1.42 (non-conventional) and 0.64 (conventional) mg GAE/g MP, respectively. For its part, the initial TPCs in the cotyledon were 4.76 and 4.30 mg GAE/g MP; after drying, they were reduced to 2.09 (non-conventional) and 1.45 (conventional) mg GAE/g MP. Based on this, we could point out that non-conventional removal (robot) contributes to preserving the phenolic compounds of cacao beans.

The degradation of polyphenols during drying is a fact that has been reported in various investigations [53]. A decrease from 86.3 to 15.0 mg GAE/g dry cacao was observed after sun drying [51], with solar drying reporting a decline from 38.18 to 23.12 mg GAE/g of dry grain. However, it should be noted that the researchers stored these grains in a vacuum after each day of drying to prevent the oxidation of phenols by oxygen.

In this study, the maximum TPC values were observed in batch III with non-conventional drying (robot), where adding the testa and cotyledon gives 10.09 and 3.55 mg GAE/g MP at the beginning and end of drying, respectively. In that sense, the TPC of 3.55 mg GAE/g MP found is more excellent than 0.8276 mg GAE/g reported by [33] but lower than that reported in other studies: [35], a content of 61.81 mg GAE/g in grains dried in the sun; [54], a range from 3 to 43 mg GAE/g in cacao bean shells; and [55], between 19.85 and 33.39 mg GAE/g of the sample in Peruvian commercial cacao beans.

In the literature, the TPC in cacao beans is very varied. Many factors could influence it, such as geographical origin [56], variety and fermentation conditions [41], drying method [57], the cultivation area, maturity, climatic conditions, harvest season, and post-harvest storage period [58], as well as the processes and parameters used during transformation [59]. On the other hand, the variations between the results found in the present research and those reported by other authors could also be attributed to the differences in the methodologies and techniques for extracting and quantifying polyphenols.

In Table 4, a clear pattern of whether the testa or the cotyledon of the grains had a higher content of phenolic compounds at the beginning is not observed. However, in all cases, after drying, the TPC was higher in the cotyledons than in the testas, likely because the testa served as a protective layer. It is also known that in cacao beans, polyphenols are stored in the pigment cells of the cotyledons [32].

Except with some exceptions, the standard deviations of the TPC averages for non-conventional drying (robot) are lower than those observed for conventional drying (Table 4). This would reinforce the previously stated idea that robot drying allows for a more uniform treatment between the grains.

The present research results indicate that the robot would help preserve the phenolic compounds in cacao beans; however, care must still be taken during transformation operations, as significant losses have been reported during roasting and alkalization [59,60]. Therefore, we emphasize that process automation in cacao post-harvest is a viable alternative to guarantee cacao quality and fair trade [61,62,63].

## 4. Conclusions

Using the robot allowed us to obtain grains with a more uniform humidity than those treated conventionally. Although it did not affect the drying time, it contributed to better preserving the phenolic compounds’ content, mainly in the cotyledons. It is beneficial at an industrial level because it adds value to the product due to the multiple benefits of these bioactive compounds. Furthermore, using the robot makes it possible to reduce the large amount of labor required during sun drying while being sustainable and economically accessible to producers in the region.

## Figures and Tables

**Figure 1 sensors-25-01520-f001:**
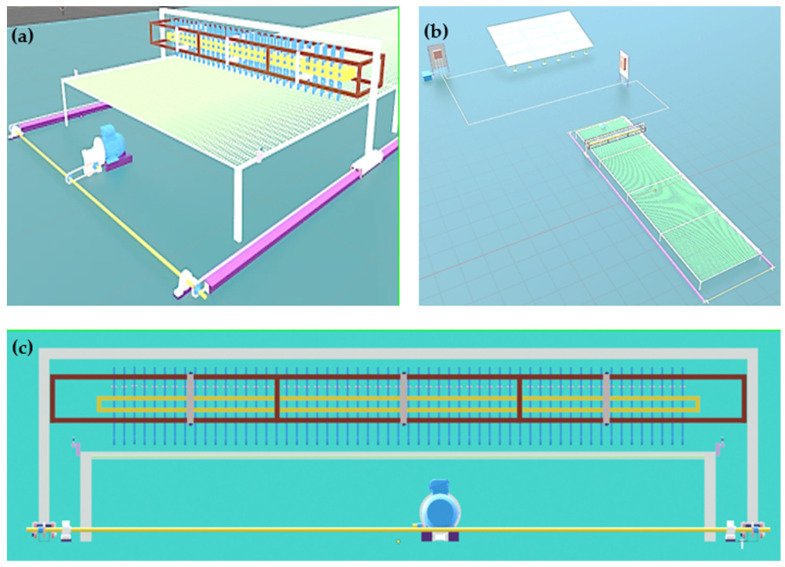
Three-dimensional modeling of the robot: (**a**) isometric view, (**b**) general scenario of the drying process, and (**c**) top view.

**Figure 2 sensors-25-01520-f002:**
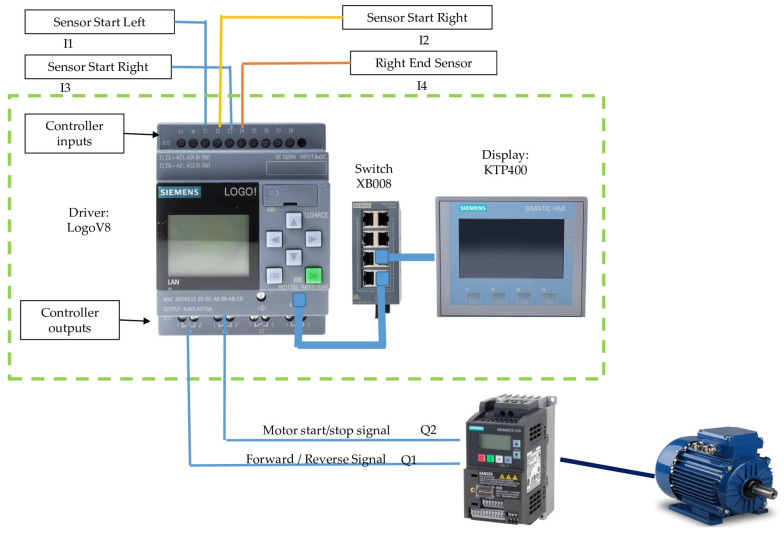
The control module of the automated removal prototype for the natural drying of the cacao bean.

**Figure 3 sensors-25-01520-f003:**
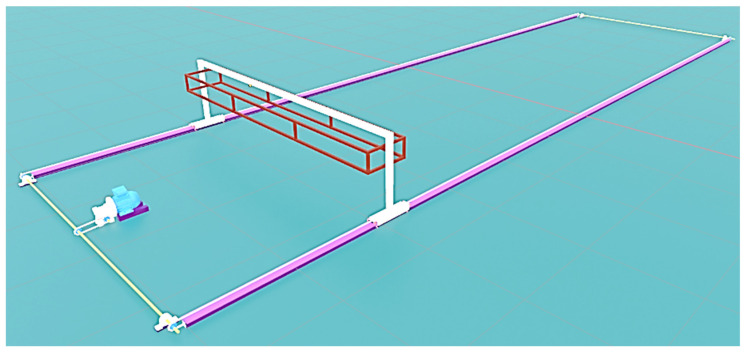
Overview of the 3D model of the automated removal system for drying cacao beans.

**Figure 4 sensors-25-01520-f004:**
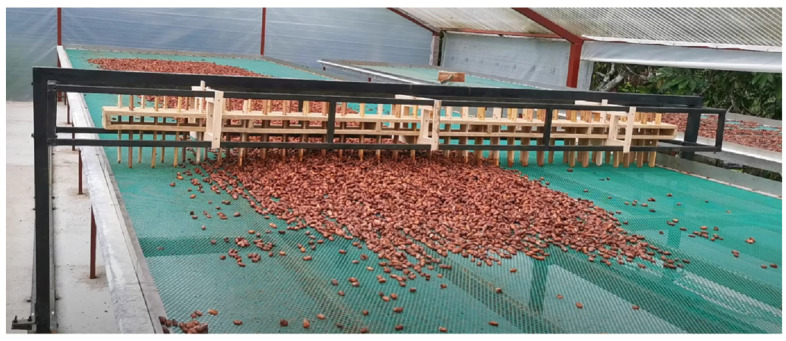
Removal module of the automated removal prototype for the natural drying of the cacao bean.

**Figure 5 sensors-25-01520-f005:**
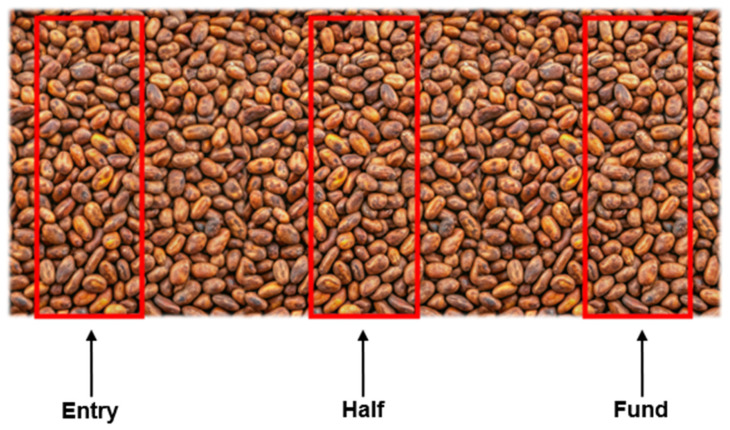
Sampling points on the cacao-bean-drying table.

**Figure 6 sensors-25-01520-f006:**
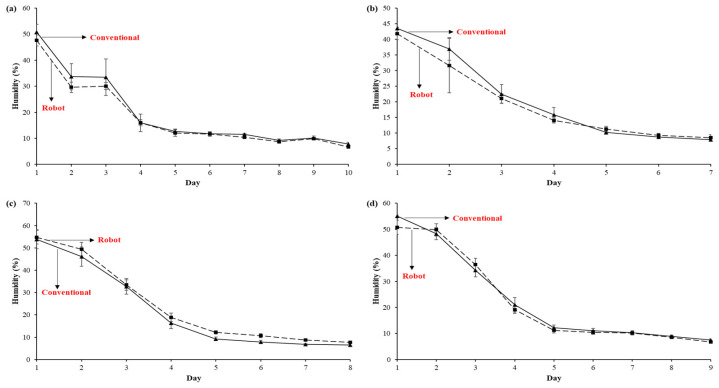
Kinetics of sun drying of cacao beans by conventional and non-conventional removal (robot): (**a**) batch I, (**b**) batch II, (**c**) batch III, and (**d**) batch IV.

**Table 1 sensors-25-01520-t001:** Hardware components used in building the robot.

Component	Manufacturer	Model
Solar panel	TrinaSolar (Changzhou, China)	TSM-455DE17M
Inverter charger	Growatt (Shenzhen, China)	SPF 5000 ES
Battery	Shoto (Yangzhou, China)	SDA10-4850
2HP frequency converter	Siemens Yangzhou, China	Sinamics V20
PLC	Siemens (Yangzhou, China)	Logo V8
Screen	Siemens Yangzhou, China	KTP400
Switch	Siemens Yangzhou, China	Scalance Xb008
2HP three-phase motor	Rainbow Yangzhou, China	MS0L-4
Switch sensor (LIMIT SWITCH)	CNC (Shanghai, China)	AZ-8108
80:1 ratio endless crown reduction box	Mark Motors (Shenzhen, China)	NMRV063
Drag chain	Honda (Minato, Japan)	428H-150L ST
Power supply	National Instruments (Austin, TX, USA)	NI PS-15

**Table 2 sensors-25-01520-t002:** Climatic conditions recorded during the removal for drying cacao beans.

**Batch**	**Inside the Drying Chamber**								
	Temperature (°C)			Moisture (%)			Light (lx)		
	Max.	Min.	Avg.	Max.	Min.	Avg.	Max.	Min.	Avg.
I	58.6	24.7	33.21	88	15	62.24	50,910	0.5	8302.61
II	52.2	24.5	31.39	90	22	69.92	50,770	0.3	7302.74
III	59.3	24.6	32.99	90	15	64.51	42,230	0.7	8027.31
IV	56.2	24.1	30.98	92	18	69.74	38,580	0.7	6350.21
**Batch**	**Outside the Drying Chamber**								
	Temperature (°C)			Moisture (%)			Wind (Km/h)		
	Max.	Min.	Avg.	Max.	Min.	Avg.	Max.	Min.	Avg.
I	36.6	23	26.84	97	43	83.32	8	0	1.259
II	33.4	23	26.23	97	61	88.50	4.8	0	0.73
III	33.9	22	26.35	97	51	85.49	6.4	0	0.791
IV	33.9	22	25.39	98	55	89.73	6.4	0	0.73

Note: Max = Maximum, Min = Minimum, and Avg = Average.

**Table 3 sensors-25-01520-t003:** Indicators of the drying and quality of cacao beans, based on the method used.

Batch	Non-Conventional (Robot)				Conventional
	Drying Time (Horas)	Energy Consumption (kW)	Removal Cycles	Grain Integrity (%)	Grain Integrity (%)
I	248	13	218	98.8	99.8
II	148	3	141	99.8	99.1
III	177	9	156	99.6	99.8
IV	192	10	151	99.3	99.6

**Table 4 sensors-25-01520-t004:** Total phenolic content (TPC, mg GAE/g MP) in cacao beans at the star and final stage of the drying process.

Batch	Grain Status	Non-Conventional (Robot)		Conventional	
		Testa	Cotyledon	Testa	Cotyledon
I	Star	5.38 ± 0.46aA^a^	4.39 ± 0.20aA^b^	2.32 ± 0.16aB^b^	3.05 ± 0.38aB^a^
	Final	0.79 ± 0.48bA^b^	2.27 ± 0.47bA^a^	0.60 ± 0.18bA^b^	2.13 ± 0.40bA^a^
II	Star	4.33 ± 0.14aA^b^	4.76 ± 0.07aA^a^	4.23 ± 0.25aA^a^	4.30 ± 0.52aA^a^
	Final	1.42 ± 0.17bA^b^	2.09 ± 0.70bA^a^	0.64 ± 0.05bB^b^	1.45 ± 0.47bB^a^
III	Star	5.56 ± 0.24aA^a^	4.53 ± 0.22aA^b^	2.94 ± 0.31aB^b^	3.77 ± 0.15aB^a^
	Final	1.28 ± 0.08bA^b^	2.27 ± 0.47bA^a^	0.57 ± 0.10bB^b^	1.58 ± 0.23bB^a^
IV	Star	3.36 ± 0.17aA^b^	4.65 ± 0.27aA^a^	3.36 ± 0.33aA^a^	3.00 ± 0.13aB^b^
	Final	0.59 ± 0.05bB^b^	1.25 ± 0.23bA^a^	0.76 ± 0.15bA^b^	1.35 ± 0.24bA^a^

Note: Data are shown as means ± standard deviations (*n* = 3). Means represented by different letters in the same column indicate significant difference.

## Data Availability

Readers who wish to obtain the database should make the request to the corresponding author.

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
