# Peer review of "Construction of an Automated Removal Robot for the Natural Drying of Cacao Beans"

_sensors, 2025, doi:10.3390/s25051520_

Round 1

Reviewer 1 Report

Comments and Suggestions for Authors

A solution involving an automated removal robot for the natural drying of cocoa beans is proposed in this paper. The integration of control, displacement, and removal modules is utilized to achieve the uniform turning of cocoa beans, providing new ideas and methods for the advancement of cocoa bean drying technology. However, certain issues remain in this paper and need further improvement.

1.  The robot described in the article is designed for specific cacao bean varieties and drying environments, but its applicability to different varieties, origins, and varying climate conditions or drying scales has not been addressed.

2.  The setting and comparison of drying cycles are not clearly explained, and the relationship between initial humidity levels and drying cycles should be clarified, including how the robot dynamically adjusts removal frequency to respond to humidity changes.

3.  The basic functions of the robot's modules are described, but detailed explanations of the mechanical design and control system (such as sensor sensitivity and control algorithms) are not provided, and mechanical design diagrams, control logic diagrams, or code are omitted, leaving the interactions among the robot's components unclear.

4.  The analysis of total phenolic content (TPC) highlights that the robot preserves phenolic compounds more effectively during drying, but the reasons for this have not been clarified, and comparisons with other drying technologies, such as forced convection drying, are absent.

5.  Figures 1-3 (robot module diagrams) lack clear illustrations of the connections between modules and detailed descriptions of each module's functions.

Author Response

Comments 1: The robot described in the article is designed for specific cacao bean varieties and drying environments, but its applicability to different varieties, origins, and varying climate conditions or drying scales has not been addressed.

Response 1: The robot was designed following the parameters of the standard cocoa procedure provided by the cooperative, ensuring its functionality in the specific conditions of the varieties and environments studied. However, to address applicability to different varieties, origins, climatic conditions and drying scales, the design includes a robotic arm with deployable tips that can be replaced to adapt to different bean dimensions. In addition, the system has the ability to adjust its removal cycles, allowing it to operate flexibly in varying conditions.

Comments 2: The setting and comparison of drying cycles are not clearly explained, and the relationship between initial humidity levels and drying cycles should be clarified, including how the robot dynamically adjusts removal frequency to respond to humidity changes.

Response 2: The observation is consistent with the suggestions of reviewer 2. We have improved the explanation of drying cycles and increased the climatic conditions to explain the relationship with robot drying.

Comments 3:  The basic functions of the robot's modules are described, but detailed explanations of the mechanical design and control system (such as sensor sensitivity and control algorithms) are not provided, and mechanical design diagrams, control logic diagrams, or code are omitted, leaving the interactions among the robot's components unclear.

Response 3: We appreciate the suggestion. We have improved the description of the robot functionalities according to the implemented modules, we also updated the figures for better understanding.

Comments 4: The analysis of total phenolic content (TPC) highlights that the robot preserves phenolic compounds more effectively during drying, but the reasons for this have not been clarified, and comparisons with other drying technologies, such as forced convection drying, are absent.

Response  4: We agree. We improve the explanation of the reasons for the effectiveness of the robot in preserving phenolic compounds and broaden the discussion by comparing with other technologies. 

Comments 5: Figures 1-3 (robot module diagrams) lack clear illustrations of the connections between modules and detailed descriptions of each module's functions.

Response 5: We agree. We have updated the images for a better understanding of the robot's functionalities.

Reviewer 2 Report

Comments and Suggestions for Authors The paper developed an automated removal robot for the natural drying of cacao beans, briefly introduced the structural composition of the robot, and finally statistically analyzed the drying effect of cacao beans. This research has strong practical significance and application value. However, it lacks in scientificity and innovation, and the amount of data is limited. In addition, the following problems still need to be addressed.

  1. In the Introduction, it is appropriate to state the cultivation status of cacao beans and the disadvantages of not drying them. However, the research status of drying technologies, drying machinery, or other aspects related to the drying of cacao beans should be added.
  2. The paper does not state or present a physical diagram of the entire device or a 3D modeling picture, making it impossible for readers to have an intuitive understanding of the robot.
  3. In Figure 1, arrows should be used to indicate each component, that is, which one is the motor, controller, power supply, etc.
  4. In Section 2.2, specify the model and manufacturer of each hardware.
  5. In Figure 2, label each sub - figure as a, b, c, d, and explain what each figure represents.
  6. Line 132: Natural drying is greatly affected by the environment. Therefore, conditions such as illumination, temperature, wind speed, and humidity should be provided. At the same time, it is necessary to explain the relationship between the drying situation, changes in substance content, and the environment.
  7. Line 254: The average energy consumption is 3 - 13kW. First, please explain what causes such a large difference in energy consumption. Second, is it possible to illustrate the traditional energy consumption situation (line 259) through data comparison?
  8. For such large - scale machinery, it is necessary to take samples and measure at different positions, compare the differences, and explain the uniformity.

Author Response

Comments 1: In the Introduction, it is appropriate to state the cultivation status of cacao beans and the disadvantages of not drying them. However, the research status of drying technologies, drying machinery, or other aspects related to the drying of cacao beans should be added.

Reponse 1: Thank you for the suggestion. In the introduction we highlighted the state of cocoa cultivation and the shortcomings of inadequate drying. We also highlight the three studies that we consider important as state of the art on cocoa drying technologies and machinery.

Comments 2:  The paper does not state or present a physical diagram of the entire device or a 3D modeling picture, making it impossible for readers to have an intuitive understanding of the robot.

Response 2: We agree with the comment. We add a subheading of 3D Modeling of the robot, and include Figure 1 with the main designs.

Comments 3:  In Figure 1, arrows should be used to indicate each component, that is, which one is the motor, controller, power supply, etc.

Response 3: We agree with the observation. We update Figure 1 (now Figure 2), for a better understanding of control module.

Comments 4: In Section 2.2, specify the model and manufacturer of each hardware.

Response 4: We add table 1 with the requested data.

Comments 5: In Figure 2, label each sub - figure as a, b, c, d, and explain what each figure represents.

Response 5: We agree. We label Figure 3 (formerly Figure 2), and explain what it represents.

Comments 6: Line 132: Natural drying is greatly affected by the environment. Therefore, conditions such as illumination, temperature, wind speed, and humidity should be provided. At the same time, it is necessary to explain the relationship between the drying situation, changes in substance content, and the environment.

Response 6: We appreciate the suggestions. We added the conditions in table 2. Also, we improved the requested explanation. 

Comments 7: Line 254: The average energy consumption is 3 - 13kW. First, please explain what causes such a large difference in energy consumption. Second, is it possible to illustrate the traditional energy consumption situation (line 259) through data comparison?

Response 7: We explain the causes of the differences in energy consumption. We also add Table 3 with data comparing the energy consumption between the robot and the conventional method. 

Comments 8: For such large - scale machinery, it is necessary to take samples and measure at different positions, compare the differences, and explain the uniformity.

Response 8: In the materials and methods section, subchapter 2.3.1. Cocoa beans, we describe the measurements of the sample in different positions by sectors. Also, the robot built in the present study has a uniform motion given the defined electromechanical structure.

Reviewer 3 Report

Comments and Suggestions for Authors

In this paper, an automated removal robot for the natural drying of cacao beans was used to replace the traditional manual method of drying cocoa beans, although the drying time was not reduced but the drying effect was better, in addition, the total phenol content was higher in the robot-dried beans, indicating better preservation of bioactive compounds, this method has a good application effect from the article.

But some problems should be explained or clarified.

1.          In section 2.2 of the assembly of the robot - prototype, the author introduces the control module, displacement module, and disassembly module, and suggests adding the overall working structure and principles of the machine to make the article more complete.

2.          The robot drying method of cocoa beans proposed in this article should explain whether the drying of cocoa beans is dried by natural ventilation or heating drying on the drying table, or other methods to avoid misunderstanding, and it is recommended to add the drying process method of cocoa beans to make the article more complete。

3.        The article mentions that the author uses solar panels instead of traditional non-renewable energy, but the article does not introduce solar panels, and should explain the parameters and principles of solar panels in Section 2.2, if only solar panels are used for power supply, then solar panels need to store enough energy to cope with the rainy season.

4.        There was even a slight increase in humidity from days 8 to 9 (Figure 2a) on line 302 Figure 2a is not marked in the article, whether there is a clerical error

5.        How does the cocoa bean drying method proposed by the authors in this study deal with the deterioration of cocoa beans caused by the long rainy season described in this paper?

6.        In section 3.2 Grain integrity, energy performance, and removal cycles of the robot (non-conventional), data on motor energy consumption and cocoa bean integrity need to be provided to enhance the integrity of the analysis of results.

Author Response

Comments 1: In section 2.2 of the assembly of the robot - prototype, the author introduces the control module, displacement module, and disassembly module, and suggests adding the overall working structure and principles of the machine to make the article more complete.

Response 1: We agree with the suggestion. We have improved the description of the three phases of robot assembly, adding 3D modeling and new figures to understand the general operating structure and its electromechanical principles.

Comments 2: The robot drying method of cocoa beans proposed in this article should explain whether the drying of cocoa beans is dried by natural ventilation or heating drying on the drying table, or other methods to avoid misunderstanding, and it is recommended to add the drying process method of cocoa beans to make the article more complete.

Response 2: We appreciate the suggestion. In the effectiveness test subchapter (2.3) we described that the drying process of cocoa beans is done by natural ventilation, also we added the climatic conditions in the drying chamber. On the other hand, we have improved the description of the robot assembly process by each module, with this the automated drying process will be better understood.

Comments 3: The article mentions that the author uses solar panels instead of traditional non-renewable energy, but the article does not introduce solar panels, and should explain the parameters and principles of solar panels in Section 2.2, if only solar panels are used for power supply, then solar panels need to store enough energy to cope with the rainy season.

Response 3: We agree with the observation. We explain the parameters and principles of operation of the solar panels that generate power for the operation of the robot. We also add Table 1 presenting the hardware components used in the construction of the robot.

Comments 4: There was even a slight increase in humidity from days 8 to 9 (Figure 2a) on line 302 Figure 2a is not marked in the article, whether there is a clerical error.

Response 4: We appreciate the clarification. We have corrected this editorial error.

Comments 5: How does the cocoa bean drying method proposed by the authors in this study deal with the deterioration of cocoa beans caused by the long rainy season described in this paper?

Response 5: The study addresses the natural drying process of cocoa beans through the implementation of a bean removal robot (non-conventional method) over the manual drying process (conventional). This process considers the use of chambers that protect the drying process from rain and provides shade. However, the chamber ensures the entry of light, as well as the entry and exit of natural wind. In this sense, the implementation of an automated non-conventional removal robot in the natural drying process seeks to minimize the deterioration of the grains due to excessive humidity, as well as the prolonged drying time.

Comments 6:  In section 3.2 Grain integrity, energy performance, and removal cycles of the robot (non-conventional), data on motor energy consumption and cocoa bean integrity need to be provided to enhance the integrity of the analysis of results.

Response 6: We appreciate the suggestion, which also coincided with reviewer 1's comment number 7. We improved subchapter 3.2, and added table 3 for a better understanding based on data.

Round 2

Reviewer 1 Report

Comments and Suggestions for Authors

The manuscript has been meticulously revised and supplemented in accordance with the comments. I believe it is suitable for publication in this journal.

Reviewer 2 Report

Comments and Suggestions for Authors

well done

Reviewer 3 Report

Comments and Suggestions for Authors

No other suggestion.